# Further Clarification of Pain Management Complexity in Radiotherapy: Insights from Modern Statistical Approaches

**DOI:** 10.3390/cancers16071407

**Published:** 2024-04-03

**Authors:** Costanza Maria Donati, Erika Galietta, Francesco Cellini, Alessia Di Rito, Maurizio Portaluri, Cristina De Tommaso, Anna Santacaterina, Consuelo Tamburella, Filippo Mammini, Rossella Di Franco, Salvatore Parisi, Sabrina Cossa, Antonella Bianculli, Pierpaolo Ziccarelli, Luigi Ziccarelli, Domenico Genovesi, Luciana Caravatta, Francesco Deodato, Gabriella Macchia, Francesco Fiorica, Giuseppe Napoli, Silvia Cammelli, Letizia Cavallini, Milly Buwenge, Romina Rossi, Marco Maltoni, Alessio Giuseppe Morganti, Savino Cilla

**Affiliations:** 1Radiation Oncology, IRCCS Azienda Ospedaliero-Universitaria di Bologna, 40138 Bologna, Italy; erika.galietta@studio.unibo.it (E.G.); filippo.mammini@studio.unibo.it (F.M.); silvia.cammelli2@unibo.it (S.C.); letiziacavallini91@gmail.com (L.C.); alessio.morganti2@unibo.it (A.G.M.); 2Department of Medical and Surgical Sciences (DIMEC), Alma Mater Studiorum University of Bologna, 40138 Bologna, Italy; mbuwenge@gmail.com; 3Dipartimento di Diagnostica per Immagini, Radioterapia Oncologica ed Ematologia, Fondazione Policlinico Universitario “A. Gemelli” IRCCS, 00168 Rome, Italy; francesco.cellini@policlinicogemelli.it (F.C.); francesco.deodato@gemellimolise.it (F.D.); 4Dipartimento Universitario Diagnostica per immagini, Radioterapia Oncologica ed Ematologia, Università Cattolica del Sacro Cuore, 00168 Rome, Italy; 5Radiotherapy Unit, IRCCS Istituto Tumori ‘Giovanni Paolo II’ Bari, 70124 Bari, Italy; aledirito@yahoo.it; 6General Hospital “Perrino”, 72100 Brindisi, Italy; portaluri@hotmail.com (M.P.); detommasocristina@gmail.com (C.D.T.); 7U.O. di Radioterapia AOOR PAPARDO PIEMONTE, 98121 Messina, Italy; anna.santacaterina@virgilio.it (A.S.); consu.universitaly@gmail.com (C.T.); 8Department of Radiation Oncology, Istituto Nazionale Tumori-IRCCS-Fodazione G. Pascale, 80131 Napoli, Italy; r.difranco@istitutotumori.na.it; 9Radioterapia Opera di S. Pio da Pietralcina, Casa Sollievo della Sofferenza, 71013 San Giovanni Rotondo, Italy; s.parisi@operapadrepio.it (S.P.); s.cossa@operapadrepio.it (S.C.); 10Medical Physics Department, IRCCS-CROB—Centro di Riferimento Oncologico della Basilica, 85028 Rionero in Vulture, Italy; 11U.O. Radioterapia Oncologica—S.O. Mariano Santo, 87100 Cosenza, Italy; pziccarelli@virgilio.it (P.Z.); lziccarelli@virgilio.it (L.Z.); 12Radiation Oncology Unit, SS Annunziata Hospital, G. D’Annunzio University of Chieti-Pescara, 66100 Chieti, Italy; d.genovesi@unich.it (D.G.); lcaravatta@hotmail.com (L.C.); 13Department of Neuroscience, Imaging and Clinical Sciences, “G. D’Annunzio” University of Chieti-Pescara, 66013 Chieti, Italy; 14Radiation Oncology Unit, Responsible Research Hospital, 86100 Campobasso, Italy; macchiagabriella@gmail.com; 15U.O.C.di Radioterapia e Medicina Nucleare, Ospedale Mater Salutis di Legnago, 37045 Verona, Italy; francesco.fiorica@aulss9.veneto.it (F.F.); napoligiuseppe.84@gmail.com (G.N.); 16Palliative Care Unit, IRCCS Istituto Romagnolo per lo Studio dei Tumori (IRST) “Dino Amadori”, 47014 Meldola, Italy; romina.rossi@irst.emr.it; 17Medical Oncology Unit, Department of Medical and Surgical Sciences (DIMEC), University of Bologna, 40138 Bologna, Italy; marcocesare.maltoni@unibo.it; 18Medical Physics Unit, Responsible Research Hospital, 86100 Campobasso, Italy; savinocilla@gmail.com

**Keywords:** observational study, multicenter, radiotherapy, pain, pain management index, least absolute shrinkage and selection operator (LASSO) algorithm, classification and regression tree (CART) analysis

## Abstract

**Simple Summary:**

This analysis of the ARISE study, a multicenter observational cohort trial, is based on a modern statistical approach, integrating the Least Absolute Shrinkage and Selection Operator algorithm and the Classification and Regression Tree analysis. The results of this study show significant shortcomings in pain management for breast cancer patients undergoing radiotherapy, particularly highlighting that younger patients and those with non-neoplastic pain, especially in southern and central Italy, experience even poorer pain management. This research underscores the urgent need for tailored pain management strategies in breast cancer patients, taking into account patient age, pain type, and geographic disparities to enhance care quality and outcomes for subjects across different regions.

**Abstract:**

Background: The primary objective of this study was to assess the adequacy of analgesic care in radiotherapy (RT) patients, with a secondary objective to identify predictive variables associated with pain management adequacy using a modern statistical approach, integrating the Least Absolute Shrinkage and Selection Operator (LASSO) algorithm and the Classification and Regression Tree (CART) analysis. Methods: This observational, multicenter cohort study involved 1387 patients reporting pain or taking analgesic drugs from 13 RT departments in Italy. The Pain Management Index (PMI) served as the measure for pain control adequacy, with a PMI score < 0 indicating suboptimal management. Patient demographics, clinical status, and treatment-related factors were examined to discern the predictors of pain management adequacy. Results: Among the analyzed cohort, 46.1% reported inadequately managed pain. Non-cancer pain origin, breast cancer diagnosis, higher ECOG Performance Status scores, younger patient age, early assessment phase, and curative treatment intent emerged as significant determinants of negative PMI from the LASSO analysis. Notably, pain management was observed to improve as RT progressed, with a greater discrepancy between cancer (33.2% with PMI < 0) and non-cancer pain (73.1% with PMI < 0). Breast cancer patients under 70 years of age with non-cancer pain had the highest rate of negative PMI at 86.5%, highlighting a potential deficiency in managing benign pain in younger patients. Conclusions: The study underscores the dynamic nature of pain management during RT, suggesting improvements over the treatment course yet revealing specific challenges in non-cancer pain management, particularly among younger breast cancer patients. The use of advanced statistical techniques for analysis stresses the importance of a multifaceted approach to pain management, one that incorporates both cancer and non-cancer pain considerations to ensure a holistic and improved quality of oncological care.

## 1. Introduction

Pain is a prevalent and debilitating symptom among cancer patients, significantly deteriorating their quality of life. Recognized by the National Cancer Institute as a symptom of primary importance, its assessment and management are crucial [1]. It is estimated that a large majority of patients with cancer may experience nociceptive or neuropathic pain during their illness trajectory [2,3].

This multidimensional syndrome not only induces physical discomfort but also exerts a profound emotional burden, leading to a decrement in the quality of life (QoL) [4,5,6,7]. The adequacy of pain control is paramount, as it is strongly associated with improved functional status and autonomy [8,9]. Despite the dissemination of comprehensive pain management guidelines and the availability of efficacious analgesic options [10,11,12,13,14], the undertreatment of pain persists.

The psychosocial impact of pain in cancer patients is profound, affecting mental health, emotional well-being, and social interactions [15]. Studies have highlighted the intricate interplay between pain and psychological distress, demonstrating how pain can exacerbate feelings of depression, anxiety, and social isolation, further diminishing the overall quality of life in cancer patients. The emotional and social ramifications of pain underscore the need for a holistic approach to pain management that addresses both the physical and psychosocial dimensions of pain [16].

Furthermore, disparities in pain management are evident, influenced by factors such as age, gender, socioeconomic status, and cultural differences. These disparities contribute to the persistent undertreatment of pain in certain patient populations, despite the established guidelines and analgesic options. Understanding these disparities is crucial for developing targeted interventions that ensure equitable pain management across diverse patient groups [17].

In addition, the integration of complementary therapies alongside conventional pain management has emerged as a promising approach for enhancing pain control in cancer patients undergoing radiotherapy. Research on complementary therapies such as acupuncture, massage, and mindfulness practices has shown potential benefits in managing pain and improving quality of life. These complementary approaches, when used in conjunction with standard pain management strategies, can contribute to more effective and holistic pain control [18].

Moreover, technological solutions for pain assessment are gaining traction, offering novel insights into the complexity of pain phenotypes in cancer patients. Emerging technologies and digital tools facilitate a more in-depth understanding of pain, enabling personalized and targeted interventions. By leveraging these technologies, healthcare professionals can enhance the precision of pain assessment and the effectiveness of pain management strategies [19]. In light of these considerations, our previous multicenter observational study sought to evaluate the management of pain in patients undergoing radiotherapy (RT) in Italian centers [19]. The study employed traditional statistical analyses to identify predictors of suboptimal pain management. Significant correlations were observed with several clinical factors, including the intent of RT, patient performance status, cancer type, and the geographical location of treatment facilities.

However, the complexity of pain phenotypes necessitates a more sophisticated analytical approach. Thus, we propose the integration of the Least Absolute Shrinkage and Selection Operator (LASSO) algorithm and the Classification and Regression Tree (CART) analysis in the present study. These advanced statistical techniques are anticipated to refine the identification of predictors by addressing multicollinearity and revealing non-linear relationships, thereby providing a comprehensive understanding of the inadequacies in pain management. Therefore, we planned a secondary analysis of the ARISE study to evaluate the effectiveness of pain relief strategies in patients receiving radiotherapy and to uncover how well pain management effectiveness correlates with various potential predictors. The insights collected from this analysis are expected to inform targeted interventions to enhance analgesic practices in RT settings.

## 2. Materials and Methods

### 2.1. Objectives

The principal aim of this study was to assess the adequacy of analgesic care in patients undergoing RT. A secondary aim was to elucidate the association between pain management adequacy and a set of potential predictive variables. These included demographic factors (gender, age), clinical status (Eastern Cooperative Oncology Group Performance Status Scale, ECOG-PS), treatment and disease-related factors (RT aim, primary tumor, and stage of disease), type of pain, and the geographical location of the RT facility.

### 2.2. Methodology

This investigation was designed as an observational, prospective, multicenter cohort study. Consent for participation was obtained in accordance with ethical standards, and the study protocol received approval from the ethics committees of all contributing centers (ARISE 327/2017/O/Oss). Eligibility for inclusion was considered for all patients presenting for medical evaluation at the participating centers within the designated study period of October to November 2019. Inclusion was independent of the timing of the visit within the patient RT course. Data were captured once per patient using a standardized collection form completed during the visit. Variables recorded included gender, age, ECOG-PS, RT aim, primary malignancy, tumor staging, and pain intensity assessed via the Numeric Rating Scale (NRS). Additional data included the analgesic regimen and categorization of pain as cancer-related, non-cancer-related, or mixed. To streamline this analysis and based on our prior findings, which demonstrated analogous correlations between cancer pain and mixed pain [19], we have combined these two patient groups for a more cohesive evaluation.

### 2.3. Inclusion and Exclusion Criteria

Criteria for inclusion were as follows: (1) diagnosis of cancer irrespective of the disease stage, primary tumor type, or RT aim, (2) undergoing treatment in RT departments, and (3) age 18 years or older. Exclusion criteria included the presence of comorbid conditions, such as psychiatric disorders or neurosensory impairments, that would preclude effective data collection or informed consent provision.

### 2.4. Outcome Measures

Pain intensity was quantified using a graded scale: 0 (NRS: 0, no pain), 1 (NRS: 1–4, mild pain), 2 (NRS: 5–6, moderate pain), and 3 (NRS: 7–10, severe pain). Concurrently, an analgesic score was assigned based on the analgesic therapy administered: 0 for no analgesics, 1 for non-opioid analgesics, 2 for “weak” opioids, and 3 for “strong” opioids. The Pain Management Index (PMI) was computed by deducting the pain score from the analgesic score, with negative PMI values indicative of insufficient analgesic treatment [20,21].

### 2.5. Variable Selection and Predictive Modeling

The study utilized the LASSO methodology alongside machine learning (ML) techniques for the dual purposes of (i) discerning robust prognostic variables, and (ii) developing and validating a predictive model utilizing the refined variable subset. LASSO functions as a sophisticated supervised learning algorithm, able to discern the influential variables associated with the outcome of interest, facilitating their retention or exclusion in the predictive model.

Subsequent to the identification of significant covariates via LASSO, the CART analysis was employed to construct the ML model. The CART algorithm operates as a non-parametric decision tree learning technique, which excels at discovering intricate patterns and elucidating inter-variable associations within voluminous datasets. The generated CART model manifests as a binary tree structure, wherein each root node denotes an input variable coupled with a corresponding threshold for bifurcation. The terminal leaves of the tree encapsulate the dependent variable, serving as the basis for prediction.

To ensure the robustness of the predictive models, a 5-fold cross-validation reiterated 100 times was performed. The performance of the model was quantitatively evaluated through the application of receiver operating characteristic (ROC) curves, complemented by the computation of the area under the curve (AUC) metric.

All statistical analysis, including Lasso and machine learning training and validation, was performed using the XLSTAT 2022.1 and glmnet statistical packages v.4.1.3 (Addinsoft, New York, NY, USA).

## 3. Results

### 3.1. Patient Characteristics

In total, 2104 individuals were recruited for participation in this investigation across 13 RT departments in Italy. Among them, 1387 patients either reported experiencing pain or were documented as receiving analgesic pharmacotherapy and had a complete dataset available. This cohort constitutes the focus of the present analysis. The demographic and clinical attributes of the patients are delineated in Table 1.

### 3.2. Pain Management Index (PMI) and Choice of Variables Included in the Predictive Model

Within the analyzed patient cohort, the prevalence of inadequately managed pain, as measured by the PMI, was 46.1%. The LASSO regression analysis identified several variables as significant determinants of a PMI score less than 0, indicative of suboptimal pain management. These variables included the nature of the pain, the geographic location of the RT center, the type of primary tumor, the ECOG PS, patient age, the timing of the evaluative assessment, the stage of the tumor, and the intended outcome of the RT.

For the construction of a clinically pragmatic predictive model, variables were selected based on the magnitude of their LASSO coefficient, specifically choosing those with a value greater than 0.2 or less than −0.2. This selection criterion resulted in the inclusion of the type of pain, the category of tumor (breast cancer versus other types), ECOG PS, patient age, timing of assessment, and the objective of RT in the final model. In an effort to enhance the applicability of the model beyond Italian medical centers, the variable representing the geographic location of the RT facility was excluded from the final predictive model.

### 3.3. Predictors of Pain Management Adequacy

The performance of the CART model was evaluated using ROC and AUC values, as reported in Figure 1. In the training and validation sets, the AUCs were 0.756 (95% CI: 0.726–0.786) and 0.742 (95% CI: 0.703–0.782), respectively, demonstrating excellent consistency.

The analysis of pain management adequacy across different patient subgroups revealed notable disparities. The incidence of inadequately managed pain was substantially higher in patients with non-cancer pain (73.1%) compared to those with cancer pain (33.2%). Within the cohort of patients with cancer pain, those receiving curative RT exhibited a higher rate of poor pain management (48.0%) relative to their counterparts undergoing palliative RT, where the rate was 27.9%. Considering patients treated with curative RT, the rate of patients with PMI < 0 was higher before RT (55.6%) compared to during RT (42.3%). Instead, in patients with cancer pain treated with palliative RT the study highlighted the influence of performance status on pain management. In fact, patients with a higher functional status (ECOG Performance Status 0–1) had a 32.7% rate of poorly managed pain, which was greater than the rate among patients with a lower functional status (ECOG Performance Status 2–4), where it was 24.9%.

Among patients with non-cancer pain, a particularly high rate of suboptimal pain management was observed in patients with breast cancer, where it reached 83.8%, in contrast to those with other types of cancer, who experienced a lower rate of 63.6%. In the latter, evaluating the timing of RT, 55.6% of patients before the commencement of RT and 42.3% during RT had inadequately managed pain. Lastly, in patients with non-cancer pain and affected by breast cancer, age appeared to be a relevant factor, with older patients (age ≥ 70 years) having a lower rate of inadequate pain management at 78.3% compared to younger patients (age < 70 years) who had a rate of 86.5%.

The findings are summarized and simplified, based on the CART binary tree structure, in a straightforward predictive model presented in Table 2.

## 4. Discussion

In our initial study published in 2022, we employed the statistical methods that were within our expertise and accessible at that time. The decision to not use advanced statistical analyses such as the LASSO algorithm and the CART was primarily due to our poor experience with these sophisticated techniques. However, recognizing the potential for a more in-depth exploration of our dataset, we employed these advanced methods in our current study. The LASSO algorithm, known for its efficiency in variable selection and regularization to enhance prediction accuracy, and CART, a decision tree technique that offers a visual representation of decision-making processes, were chosen with the expectation of deriving more nuanced insights. In fact, unlike traditional multivariate analysis, which simply identifies parameters correlated with the quality of pain management, LASSO and CART allow for the evaluation of hierarchical relationships among significant factors, thus providing a robust predictive model.

In this large multicenter study, which assessed more than a thousand individuals undergoing RT, it was found that 46.1% of the patients experienced inadequate pain management, as indicated by a PMI of less than 0. The insufficiency in analgesic treatment was notably associated with several factors, including the non-cancer origin of pain, the type of tumor (with breast cancer compared to other types), the better ECOG PS scores, the younger age of the patient, the earlier point in time when the assessment was made, and the curative goal of RT.

Previous research has shown that pain not related to cancer, such as that from other health issues, is often linked to a higher occurrence of negative PMI scores [22,23]. These findings suggest that pain from non-cancer sources can be overlooked in treatment management. The present analysis adds relevant information to these findings. In fact, it demonstrates that the cause of pain is actually the most significant factor in how well pain is managed, with a notable difference in the adequacy of pain control between patients with pain from cancer (33.2%) and pain from other conditions (73.1%). This highlights the need for a comprehensive approach in oncological practice where physicians consider all aspects of a patient’s health, not just cancer-related issues [24].

Furthermore, in this study, it was found that a negative PMI was more prevalent among patients receiving curative RT as opposed to those with palliative treatment intentions. This aligns with observations made by Fujii et al., who noted a higher incidence of negative PMI in patients undergoing adjuvant chemotherapy compared to those receiving it for advanced diseases [25]. Additionally, our study revealed that patients with an ECOG PS of 0–1 had a higher frequency of negative PMI compared to those with a status of 2–4, confirming the results of previous studies [25,26]. Notably, the current analysis enriches these findings by highlighting that the improved pain management observed in patients with better ECOG scores was recorded only in patients undergoing palliative RT. This result suggests that within the palliative care framework, healthcare providers may exhibit heightened vigilance toward the overall condition of the patient, especially in managing pain.

Moreover, our previous findings highlighted a higher prevalence of negative PMI scores among breast cancer patients as compared to those with other cancer types (60.5% vs. 30.9–49.4%) [19], challenging the notion that this discrepancy is solely based on gender. In fact, gender was not a significant predictor of PMI in our analysis, and the trend of higher negative PMI persisted even when compared to other female-specific cancers such as endometrial and cervical cancers [19]. Other studies have also reported a link between breast cancer and higher rates of negative PMI [23,26,27]. We hypothesized that several factors might predispose breast cancer patients to poor pain management: they often have a good clinical condition (ECOG-PS 0–1), their pain is likely due to non-cancer causes like post-surgical effects, and they are typically receiving adjuvant treatment with curative intent.

The current analysis advances these initial hypotheses by clarifying the complexity within the patient groups. Notably, the distinction between breast cancer and other cancers in terms of PMI was particularly evident in patients with non-cancer pain, providing concrete data to support our previous suppositions about the significant role of non-cancer pain in these patients’ negative PMI scores.

Additionally, our research revealed another interesting pattern within the breast cancer population. The highest rate of negative PMI scores (86.5%) was observed in breast cancer patients under 70 years of age who reported non-neoplastic pain. This outcome suggests that younger patients may be especially vulnerable to inadequate management of benign pain and it indicates a need for heightened awareness and more tailored pain management strategies for younger breast cancer patients.

Another original finding of our analysis is the enhancement in pain management as patients progress through RT treatment. This is a positive development and suggests that the routine, daily interaction patients have with the RT department plays a relevant role in enabling physicians to better recognize and manage pain. One possible confirmation of this is that the improvement in pain management appears more pronounced for those undergoing curative RT, which typically involves a longer treatment duration, as opposed to palliative RT, which is often shorter. However, an alternative hypothesis could be that pain management is more adequate in the palliative RT setting, which might obscure the observed improvement during treatment.

The situation is less straightforward when it comes to non-cancer pain, particularly in breast cancer patients. Unlike patients with other types of tumors, breast cancer patients usually undergo RT after surgery, and their pain is often associated with the surgical site. This type of pain is known to be persistent over time and may not show the same degree of improvement during RT as pain from other benign causes. Therefore, while the general trend indicates an improvement in pain management during RT, the specific case of postoperative pain in breast cancer patients may require additional targeted strategies to achieve similar outcomes.

Literature suggests that around 43% of cancer patients have a negative PMI, although there is evidence of a decreasing trend in recent years [28]. Our findings, showing 46.1% of patients with a negative PMI, align with this reported range [14] and are consistent with other PMI-focused studies that have found similar rates (39.7–53.0%) [25,27,29]. However, some studies have reported much higher rates of inadequate pain management (77–83%) [22,30], potentially due to factors such as geographic location or patient demographics, such as younger patients in better clinical condition who have been shown to have higher negative PMI rates [30].

On the other hand, several studies have reported lower rates of negative PMI (4–33.3%) [23,26,31,32,33,34,35], which in some instances are attributed to the focus on patients receiving palliative care [26,31,33]. This is supported by our analysis and that of Fujii et al. [25], which both suggest that pain management tends to be more effective in palliative care settings. Additionally, the enhanced management of pain in some reports may reflect specialized care in supportive or palliative care units [32,34,35], underscoring the potential benefits of integrated, patient-focused treatment approaches.

This study has several limitations. Primarily, it focused solely on pain management without assessing how this impacts the overall QoL. Additionally, the PMI, our chosen metric, has its drawbacks, such as its outdated categorization of opioids and its questionable link with QoL outcomes [21]. Specifically, a PMI score less than 0 does not consistently align with a patient’s expressed need for better pain management, although lower PMI scores often correspond with more frequent reports of pain disrupting daily activities [27,35]. Moreover, PMI typically reflects prescribed rather than actual medication usage [20,21,36,37]. However, our study mitigated this by collecting data based on patients’ reported consumption, not just prescriptions. Furthermore, PMI assumes adequate treatment in all patients taking strong opioids, irrespective of medication type, dosage, and actual pain relief experienced, which is not always the case. Nevertheless, despite these drawbacks, PMI is widely used due to its association with pain treatment quality and its straightforward calculation and data-gathering process [38]. Lastly, our study did not distinguish between types of pain, such as nociceptive, neuropathic, or mixed, precluding analysis of how these classifications might influence pain management adequacy.

Despite the aforementioned limitations, our study possesses several notable strengths. The large number of patients from multiple centers across Italy means our results are likely to apply to a wide range of RT contexts. The use of modern statistical methods, like LASSO and CART, has given us a clearer picture of the complex factors that affect how pain is managed in cancer care. Additionally, by focusing on the actual medications that patients take, rather than what is simply prescribed, we have a more realistic view of pain management in practice. Together, these aspects of our study provide valuable information that can help improve how pain is treated in patients undergoing RT.

## 5. Conclusions

In our large study, we identified that 46.1% of patients undergoing RT had inadequately managed pain. This suboptimal pain management was associated with a range of factors, such as non-cancer pain origins, breast cancer diagnosis, higher ECOG performance status, younger patient age, initial assessment phase, and curative treatment intent.

A significant original finding is that pain management improved during the course of RT, particularly for those with cancer-related pain undergoing longer curative treatment. In addition, we observed that patients with worse ECOG scores often received better pain management, a phenomenon specifically noted in the palliative care setting. Another novel finding from our research is the particularly high rate of negative PMI scores (86.5%) among breast cancer patients under 70 years of age with non-cancer pain, pointing to a potential oversight in addressing benign pain in younger breast cancer patients.

These insights underline the need for a comprehensive, patient-centric approach in oncological care, where both cancer and non-cancer pain are carefully managed to improve overall pain relief.

## Figures and Tables

**Figure 1 cancers-16-01407-f001:**
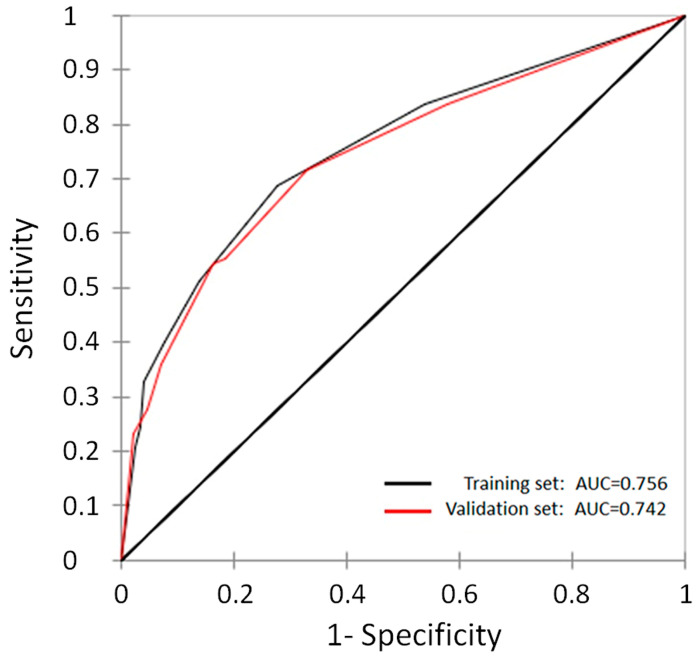
ROC curves and AUC values obtained by the CART model for the training (black solid line) and validation (red solid line) sets, respectively.

**Table 1 cancers-16-01407-t001:** Patients’ Characteristics.

		Number	(%)
** *Gender* **			
Male		657	47.4
Female		730	52.6
** *Age, years* **			
≤70		855	61.6
71–80		377	27.2
>80		155	11.2
** *ECOG-PS* **			
0–1		852	61.4
2		343	24.7
3		163	11.8
4		29	2.1
** *Aim of treatment* **			
Curative		656	47.3
Palliative		731	52.7
** *Primary tumor* **			
Breast		426	30.7
Prostate		149	10.7
Gastrointestinal		137	10.0
Endometrial/cervical		74	5.3
Lung		197	14.2
Head and neck		122	8.8
Others		282	20.3
** *Tumor stage* **			
Metastatic		759	54.7
Non-metastatic		628	45.3
** *Type of pain* **			
Cancer pain or mixed pain		941	67.9
Non-cancer pain		446	32.1
** *Pain score* **			
(NRS: 0)	0	34	2.5
(NRS: 1–4)	1	591	42.6
(NRS: 5–6)	2	509	36.7
(NRS: 7–10)	3	253	18.2
** *Analgesic score* **			
(No therapy)	0	327	23.6
(Analgesics)	1	572	41.2
(Weak opioids)	2	197	14.2
(Strong opioids)	3	291	21.0
** *Location of the radiotherapy center* **			
Northern Italy		272	19.6
Central Italy		168	12.1
Southern Italy		947	68.3
** *Timing of visit* **			
During therapy		748	54.0
End of therapy		639	46.0

Legend: ECOG-PS: Eastern Cooperative Oncology Group Performance Status Scale; NRS: Numeric Rating Scale.

**Table 2 cancers-16-01407-t002:** Predictive model for inadequate pain management: red numbers represent the proportion of patients with inadequate pain management (PMI < 0), while the figures in brackets represent the total number of patients within each respective group (RT: radiotherapy; ECOG: Eastern Cooperative Oncology Group Performance Status).

**ALL PATIENTS** 46.1 (1387)							
**CANCER PAIN** 33.2 (941)				**NON-CANCER PAIN** 73.1 (446)			
**AIM OF TREATMENT**				**PRIMARY TUMOR**			
Curative RT 48.0 (250)		Palliative RT 28.0 (691)		Breast cancer 83.8 (210)		Other cancers 63.6 (236)	
Before RT	During RT	ECOG-PS 0–1	ECOG-PS 2–4	Age < 70 years	Age = 70 years	Before RT	During RT
55.6 (108)	42.3 (142)	33.0 (269)	25.0 (422)	86.5 (141)	78.3 (69)	69.0 (106)	59.0 (130)

## Data Availability

Data supporting reported results can be found at the Radiotherapy Unit—A.G. Morganti of the IRCCS Azienda Ospedaliero-Universitaria di Bologna.

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
