# Peer review of "Further Clarification of Pain Management Complexity in Radiotherapy: Insights from Modern Statistical Approaches"

_cancers, 2024, doi:10.3390/cancers16071407_

Round 1
Reviewer 1 Report
Comments and Suggestions for Authors
Authors proposed a paper entitled “Further clarification of pain management complexity in radiotherapy: insights from modern statistical approaches” for the publication in Cancers.
The addition of an abbreviation list would be well appreciated.
The introduction is well-structured, informative, and logically leads to the proposed research methodology. It successfully engages the reader by conveying the significance of the research topic and the innovative approach you plan to take in analyzing the data.
However, expanding the introduction could involve providing additional context, statistical data, or emphasizing the broader implications of your research.
These are the related topics that authors could expand in their introduction:
-Psychosocial Impact of Pain in Cancer Patients: Explore the psychosocial dimensions of pain, examining its impact on mental health, emotional well-being, and social interactions. Discuss studies that highlight the intricate interplay between pain, psychological distress, and overall quality of life in cancer patients.
-Disparities in Pain Management: Investigate any existing disparities in pain management, considering factors such as age, gender, socioeconomic status, or cultural differences. Discuss how these disparities might contribute to the persistent undertreatment of pain, despite the availability of guidelines and analgesic options.
-Integration of Complementary Therapies: Discuss research on the integration of complementary therapies alongside conventional pain management in cancer patients undergoing radiotherapy. Explore whether complementary approaches contribute to more effective and holistic pain control.
-Technological Solutions for Pain Assessment: Touch upon emerging technologies or digital solutions used for pain assessment in cancer patients. Discuss how these tools may offer additional insights into the complexity of pain phenotypes and contribute to more targeted interventions.
Objectives should be declared at the end of the introduction section.
Line 235. I would not call this paragraph as “3.2. Figures and Tables”
Figure 1. Please improve the focus of this diagram. Moreover, the comma should be substituted by dots, in decimal separation.
The caption of figure should be set after the figure itself. Please check with this journal guidelines.
Figure 2 is not a figure, but a table. Please transform this figure into a table.
Line 293. “we found that a negative” I would not use personal form, but impersonal.
Line 385. “In our large study we identified” according to previous comment, please rephrase this sentence.
Comments on the Quality of English LanguageA quite good use of English
Author Response
Please see the attachement

Reviewer 2 Report
Comments and Suggestions for Authors
Thank you for permiting me to review this manuscript
THis manuscript appears to be new results from a previously published study ref 15 using novel statistical method
since the first study is not too old (2022) , the authors should explain why they did not use it in the first study
The authors should elaborate more details about the new statistical method and give some explanations for newly used test such as Least Absolute Shrinkage and Selection Operator (LASSO) algorithm and the Classification and Regression Tree (CART) and what were the expectations ? and clearly explain why this would bring more relevance in comparison to the study published earlier
Since a large subgroup of patients are breast cancer patients the contribution of surgery and anesthesia techn ique to chronic pain could have been investigated
It appears to me that novel statistical techniques explored other data than data explored in the first study may be a title such as part 2 would have been more rational than novel statistical approach
Comments on the Quality of English Languagemay be a native english speaker could improve some phrases as
Round 2
Reviewer 1 Report
Comments and Suggestions for Authors
Authors provider a new versione of their paper.
This new version was significantly modified by authors, that responded to my issues point by point. For these reasons, the paper deserves to be published in present form.
Reviewer 2 Report
Comments and Suggestions for Authors
The authors have responded to my queries adequately